# Predictive Role of Neutrophil-Percentage-to-Albumin Ratio (NPAR) in Nonalcoholic Fatty Liver Disease and Advanced Liver Fibrosis in Nondiabetic US Adults: Evidence from NHANES 2017–2018

**DOI:** 10.3390/nu15081892

**Published:** 2023-04-14

**Authors:** Chi-Feng Liu, Li-Wei Chien

**Affiliations:** 1School of Nursing, National Taipei University of Nursing and Health Science, Taipei 112, Taiwan; 2Department of Obstetrics and Gynecology, School of Medicine, College of Medicine, Taipei Medical University, Taipei 112, Taiwan; 3Department of Obstetrics and Gynecology, Taipei Medical University Hospital, Taipei 112, Taiwan

**Keywords:** liver fibrosis, National Health and Nutrition Examination Survey (NHANES), neutrophil-percentage-to-albumin ratio (NPAR), neutrophil-to-lymphocyte ratio (NLR), nonalcoholic fatty liver disease (NALFD)

## Abstract

Nonalcoholic fatty liver disease (NAFLD) is highly prevalent globally and includes chronic liver diseases ranging from simple steatosis to nonalcoholic steatohepatitis (NASH). The neutrophil-to-albumin ratio (NPAR) is a cost-effective, readily available biomarker of inflammation used to assess cancer and cardiovascular disease prognosis, and it may be of predictive value in NAFLD. This study was to evaluate the associations between the NPAR, the neutrophil-to-lymphocyte ratio (NLR), and the presence of NAFLD or advanced liver fibrosis, and to assess the predictive value of the NPAR in NAFLD in a nationally representative database. This population-based, cross-sectional, retrospective study analyzed the secondary data of adults with NAFLD or advanced liver fibrosis extracted from the National Health and Nutrition Examination Survey (NHANES) database 2017–2018. NHANES participants with complete information of vibration-controlled transient elastography (VCTE) and controlled attenuation parameter (CAP) were enrolled. A logistic regression analysis was used to determine the associations between the variables in the participants with and without NAFLD or advanced liver fibrosis. The mean values of the lymphocyte counts, neutrophil counts, NPAR, aspartate aminotransaminase (AST), alanine aminotransaminase (ALT), total cholesterol, triglycerides, and HbA1c were significantly higher in the participants with NAFLD than in those without NAFLD or advanced liver fibrosis. The mean blood albumin levels of the subjects without NAFLD or advancing fibrosis were considerably greater than those of the individuals with these conditions. The mean values of the NLR, NPAR, AST, ALT, triglycerides, lymphocyte count, neutrophil count, and HbA1c were significantly higher in patients with advanced fibrosis than in those without advanced fibrosis. A multivariate analysis showed that per unit increases in both the NLR and NPAR were significantly associated with an increased risk of developing NAFLD, while neither the NLR nor NPAR was significantly associated with higher odds of advanced fibrosis. In conclusion, the novel biomarker NPAR demonstrates a good association with NAFLD, along with participants’ clinical characteristics, in a nationwide population. The NPAR may serve as a biomarker for NAFLD and help clinicians refine the diagnosis and treatment of chronic liver disease.

## 1. Introduction

From simple steatosis to nonalcoholic steatohepatitis (NASH), these are all symptoms of nonalcoholic fatty liver disease (NAFLD). The global prevalence of NAFLD is 25% [1], and the prevalence in the US population is approximately ≥ 34.0%, among whom 3% to 5% are estimated to have the more severe form of progressive NASH [2]. A recent meta-analysis reported a significant increase in the prevalence of NAFLD in Asia, estimated to be 29.6% currently, with considerable variation shown between countries [3]. NAFLD is characterized by abnormal lipid deposition in the liver, hepatocellular injury, necroinflammation, and the rapid progression of fibrosis [4]. NAFLD may further develop into cirrhosis and increase the risk of hepatocellular carcinoma [5]. Patients with NAFLD are at higher risk of dying from liver disease, cardiovascular disease, and malignancy [6,7].

However, the prediction of clinical outcomes in NAFLD remains challenging because the factors underlying its progression have not been fully identified [8]. NAFLD is considered to be a manifestation of metabolic syndrome, with clinical outcomes strongly associated with obesity, type 2 diabetes, dyslipidemia, and endothelial dysfunction [9]. NAFLD occurs in up to 75–100% of obese individuals [10]. About 20% of individuals with NAFLD develop NASH associated with cirrhosis, portal hypertension, and hepatocellular carcinoma, but the mechanism underlying the progression from NAFLD to NASH remains unclear. Compared with the incidence in other liver diseases, a large proportion (35–50%) of hepatocellular carcinoma in NASH develops in patients with cirrhosis and before routine cancer screening [11]. Rapidly advanced fibrosis is a hallmark of NASH disease progression, and reducing fibrosis is an important goal of therapy for all liver disease. 

The early detection and evaluation of NAFLD and liver fibrosis are crucial for tracking disease development and choosing appropriate therapeutic approaches for afflicted patients [12]. For a clinical diagnosis and the grading of liver fibrosis in chronic hepatitis, a liver biopsy is the gold standard. However, a liver biopsy is intrusive, rather expensive, and fraught with danger. Therefore, being able to identify the NAFLD and severity of liver fibrosis noninvasively would be of great benefit. A previous animal study suggested that the hepatocyte inflammasome may be an important link between hepatocyte death and the stimulation of fibrogenesis in NASH, and it may be a noninvasive indicator of inflammation [13]. NASH patients have higher levels of inflammatory cytokines, which may cause chronic inflammation and promote disease progression. Systemic inflammation also has a recognized role in the pathogenesis of advanced liver cirrhosis [14]. The NLR is an indicator of inflammation, using two values—neutrophil and lymphocyte counts—easily obtained from routine blood tests [15]. Therefore, NLR can be used as a marker to reflect the NAFLD and severity of liver fibrosis.

Similarly, the NPAR is an effective biomarker that uses neutrophil counts and albumin values to provide a cost-effective and readily available indicator of systemic inflammation. Acute kidney injury, cardiogenic shock, myocardial infarction, and cancer patients can all be predicted with the help of the NPAR, according to earlier research [16,17,18]. To our knowledge, no study has looked at the predictive value of the NPAR in NAFLD or advanced liver fibrosis. Therefore, this study aimed to evaluate the associations between the NLR, the NPAR, and the presence of NAFLD or advanced liver fibrosis, and to assess the predictive value of the NPAR in NAFLD in a large, nationally representative population. 

## 2. Methods 

### 2.1. Study Design and Data Source

This study is a retrospective, cross-sectional, population-based analysis of secondary data extracted from the 2017–2018 National Health and Nutrition Examination Survey (NHANES) database, which is collected and maintained by the National Center for Health Statistics (NCHS), a division of the US Centers for Disease Control and Prevention (CDC). The NHANES survey is designed to assess the health and nutritional status of non-institutionalized individuals across the US, using a complex, multistage design that enables the collection and analysis of representative data at a national level. Participants undergo an extensive evaluation process, including a household interview and examination at a mobile examination center (MEC) that comprises physical examination, specialized measurements, and laboratory tests. As the data are released for research purposes and researchers are granted permission to use the data by the NCHS, the NHANES database provides a reliable and comprehensive evaluation of the population and can be considered a population-level assessment [19]. 

### 2.2. Study Population

The present study extracted data from the released 2017–2018 cycle of the NHANES database. Adults aged ≥ 18 years old with results of vibration-controlled transient elastography (VCTE), an indicator of liver stiffness and advanced liver fibrosis that is available only in the 2017–2018 NHANES study cycle, were included. Pregnant women, participants with history of excessive alcohol consumption (defined as >21 standard drinks per week in males; >14 standard drinks per week in females), positive serological markers for hepatitis B or C virus, physician-diagnosed hepatitis B or C, history of malignancy, and end-stage renal disease (ESRD) (defined by an estimated glomerular filtration rate (eGFR) < 15), an AST or ALT > 500 IU/L, or no data on AST or ALT were excluded. 

### 2.3. Ethical Considerations

The NHANES program was subjected to review and approval by the NCHS Research Ethics Review Board, and all participants in the survey provided signed informed consent. As the NHANES data released by the NCHS are de-identified and anonymous during data analysis, performing secondary analyses on the data does not require any additional ethical approval or informed consent. The NCHS Research Ethics Review Board’s approval can be accessed on the NHANES website (https://www.cdc.gov/nchs/nhanes/irba98.htm) (accessed on 19 December 2022).

### 2.4. Study Variables

#### 2.4.1. Measurement of NAFLD and Advanced Liver Fibrosis

Liver stiffness was evaluated using the controlled attenuation parameter (CAP) feature of the Fibroscan model (Echosens North America, Waltham, MA, USA), which reflected liver fibrosis accurately [20]. Liver steatosis (AUROC: 0.96) was detected using the noninvasive VCTE, as previously described [20,21]. In the NHANES data set, 4266 participants had undergone VCTE, using FibroScan model 502 V2 Touch (Echosens, North America) with a medium (M) or extra-large (XL) wand (probe) in the NHANES MEC. In particular, in accordance with previous studies, NAFLD was defined as having a CAP ≥ 285 dB/m [22], and advanced liver fibrosis was defined as having a VCTE ≥ 12 kPa [23].

#### 2.4.2. Measurement of Indicators of NPAR and NLR 

Hematologic parameters were measured following the NHANES CBC Profile using the Beckman Coulter Automated Hematology Analyzer DxH 900 (Beckman-Coulter, Brea, CA, USA), which performs red and white cell counts, hemoglobin, hematocrit, and red blood cell indices. The Coulter VCS system is used for the WBC differential. The Beckman Coulter Analyzer system counts and sizes cells using an automatic dilution and mixing system for sample processing, and a single beam photometer for hemoglobinometry. The NLR was determined for each participant by dividing the total absolute neutrophil count by the total absolute lymphocyte count in the WBC. NPAR was calculated using the same blood sample and the following formula: Neutrophil percentage (in total WBC count) (%) × 100/Albumin (g/dL). 

#### 2.4.3. Covariates

Trained NHANES interviewers obtained demographic data, including age, sex, race, poverty-to-income ratio, and educational level, from in-person interviews, using the Family and Sample Person Demographics questionnaires and the Computer-Assisted Personal Interviewing (CAPI) system (Confirmit Corp., New York, NY, USA). The collected data were weighted in accordance with the NHANES protocol. Body mass index (BMI) was calculated from the NHANES examination measurements, which were obtained as body weight (in kilograms) divided by height (in meters squared). 

Identification of participants with diabetes mellitus (DM) was conducted through at least one of the positive responses to questions “Are you taking insulin?”, “Did a doctor tell you that you have diabetes?”, “Do you take pills to lower blood sugar?”, or having an HbA1c ≥ 6.5%, and a fasting glucose ≥ 126 mg/dL in the laboratory data set [24]. 

Hypertension was defined by those who responded “yes” to the questions of “Were you told on two or more different visits that you had hypertension, also called high blood pressure?” or “Because of your (high blood pressure/hypertension), have you ever been told to… take prescribed medicine?”, or with an average of three consecutive measures of systolic blood pressure ≥ 140 mmHg, or with an average of three consecutive measures of diastolic blood pressure ≥ 90 mmHg. History of CVD was defined through positive responses to questions about physician diagnoses of myocardial infarction, angina, coronary heart disease, congestive heart failure, or stroke. Chronic kidney disease (CKD) was defined as an estimated GFR < 60 mL/min/1.73 m^2^. The estimated glomerular filtration rate (GFR) was calculated using the re-calibrated serum creatinine and the 4-variable Modification of Diet in Renal Disease (MDRD) Study equation. In this study, the IDMS-traceable MDRD Study equation that employs standardized creatinine was utilized: GFR = 175 × (standardized serum creatinine) − 1.154 × (age) − 0.203 × 0.742 (if the subject is female) × 1.212 (if the subject is African American). Smoking status was categorized as non-smoker, former smoker, or current smoker based on the following criteria: participants with a lifetime smoking history of less than 100 cigarettes were classified as non-smokers, those with a lifetime smoking history of more than 100 cigarettes but not currently smoking were classified as former smokers, and those with a lifetime smoking history of more than 100 cigarettes who responded affirmatively to the question “Do you smoke now?” were classified as current smokers. Laboratory profiles, including aspartate aminotransferase (ASL), alanine aminotransferase (ALT), serum total cholesterol, triglycerides, and HbA1c, were obtained from the NHANES laboratory data files. AST and ALT values were measured using the kinetic rate method on the Roche Cobas 6000 (c501 module) Analyzer (Roche Diagnostics, Indianapolis, IN, USA).

### 2.5. Statistical Analysis

Due to the complex sampling design of the NHANES database, all statistical analyses were conducted using SAS survey analysis statements to produce nationally representative estimates (SAS Institute Inc., Cary, NC, USA). We employed weighted samples, stratum, and cluster of the NHANES database, and the SURVEY procedure in SAS was utilized to compute the weighted population of this study. Continuous variables were reported as weighted mean and standard error, while categorical variables were reported as unweighted numbers and weighted proportions. Weighted samples were generated in accordance with the analytical guidelines published by the NCHS. We utilized the SURVEYLOGISTIC statement to perform logistic regression analysis and examine the associations between NLR, NPAR, FLI, and the presence of NAFLD and advanced liver fibrosis. Multivariate regression analysis was adjusted for potential confounding factors found to be significant in univariate regression, including age (in years), sex, race, BMI, smoking status, and comorbid conditions (DM, hypertension and history of CVD). Associations between NPAR, FLI, and NAFLD and advanced liver fibrosis were evaluated further by checking the diagnostic performance of NPAR, FLI for NAFLD in subjects without DM by using the area under the receiver operating characteristics (AUROC). A two-sided *p*-value of <0.05 was regarded as statistical significance.

## 3. Results

### 3.1. Study Cohort Selection

The selection of the study cohort is shown in a flow diagram in Figure 1. There were 9254 identified NHANES participants who underwent interviews throughout the 2017–2018 period. Of these, 5119 subjects with VCTE and CAP data and a minimum age of 18 were initially chosen. Finally, 3991 subjects were included in the final cohort as the analytic samples after excluding 356 subjects with missing data on AST or ALT, 136 subjects with excessive alcohol consumption, 71 patients with positive serological markers for hepatitis B or C virus, 93 patients diagnosed with hepatitis B or C, 463 patients having a history of cancer, and 9 subjects having ESRD. This sample size represented a population of 182,233,561 US adults after weighting (Figure 1).

### 3.2. Clinical Characteristics of the Study Sample

The characteristics of the study sample are summarized in Table 1. The participants’ mean age was 45.3 years, with 48.8% males and 51.2% females. Most participants were non-Hispanic White (60.4%) and never smokers (60.9%). The mean BMI was 29.8 kg/m^2^. The means for the lymphocyte count, neutrophil count, NPAR, FLI, AST, ALT, total cholesterol, triglycerides, and HbA1c in the subjects without NAFLD were significantly lower than those with NAFLD. The mean serum albumin level in the subjects without NAFLD was significantly higher than that in those with NAFLD. In addition, the mean NPAR, FLI, AST, ALT, triglycerides, neutrophil count, and HbA1c in the subjects without advanced liver fibrosis was significantly lower than that in those with advanced fibrosis. The mean albumin in those without advanced fibrosis was significantly higher than in those with advanced fibrosis. (Table 1).

### 3.3. Associations between NPAR, NLR, FLI, the Presence of NAFLD and Advanced Liver Fibrosis

The associations between the NPAR, NLR, FLI, prevalent NAFLD and advanced liver fibrosis are summarized in Table 2. In a multivariable analysis, the per unit increases in the NLR and NPAR were both significantly associated with a decreased risk of NAFLD (adjusted odds ratio (aOR) = 0.88, 95% confidence interval (CI): 0.77–1.00; aOR = 0.93; 95% CI: 0.88–0.99, respectively). When regarded as quartiles, compared with the lowest quartile (Q1), the subjects in the highest quartile of the NPAR (Q4) were significantly less likely to have NAFLD (aOR = 0.64; 95% CI: 0.43–0.95). The per unit increases in the FLI were significantly associated with an increased risk of NAFLD (aOR = 1.04; 95% CI: 1.03–1.05). Compared with the lowest quartile (Q1), the subjects in the higher quartile of the FLI were significantly associated with the odds of NAFLD (Q3 vs. Q1: aOR = 11.07; 95% CI: 5.27–23.26; Q4 vs. Q1: aOR = 33.14; 95% CI: 14.35–76.51). In addition, the per unit increases in the NPAR were significantly associated with an increased risk of advanced fibrosis (aOR = 0.89; 95% CI: 0.79–0.99). Instead, the per unit increases in the NPAR were significantly associated with an increased risk of advanced fibrosis (aOR = 1.03; 95% CI: 1.00–1.05) (Table 2).

### 3.4. Associations between NPAR and Presence of NAFLD and Advanced Liver Fibrosis Stratified by DM Status

The participants were further stratified by whether or not they had DM, documented in Table 3. After adjustments, among the individuals with DM, it was found that the NPAR was not significantly associated with NAFLD. However, an increased FLI was significantly and independently associated with the odds of developing NAFLD (aOR = 1.04; 95% CI: 1.02–1.06), where the aOR for the FLI Q2, Q3, and Q4 was 3.29 (95% CI: 1.02–10.60), 6.01 (95% CI: 1.48–24.44), and 9.23 (95% CI: 1.39–61.30) versus Q1, respectively. Among the individuals without DM, the NPAR was significantly associated with NAFLD (aOR = 0.91; 95% CI: 0.85–0.97), where the aOR for the NPAR Q4 was 0.56 (95% CI: 0.36–0.88) versus Q1, respectively. Furthermore, the per unit increases in the FLI were significantly associated with an increased risk of advanced fibrosis (aOR = 1.04; 95% CI: 1.03–1.05). Compared with the lowest quartile (Q1), the subjects in the higher quartile of the FLI (Q2, Q3, and Q4) were significantly more likely to have NAFLD (Q2 vs. Q1: aOR = 7.00; 95% CI: 3.43–14.29; Q3 vs. Q1: aOR = 12.09; 95% CI: 5.48–26.65; Q4 vs. Q1: aOR = 41.01; 95% CI: 16.30–103.17). In addition, the NPAR and FLI were not significantly associated with the odds of advanced fibrosis in individuals with or without DM (Table 3).

### 3.5. ROC Analysis of the Predictive Value of NPAR for NAFLD in Nondiabetic Individuals

The ROC analysis was conducted to evaluate the discriminative power of the NPAR and FLI for NAFLD in individuals without DM. The results showed that the AUROC of the NPAR (combined with demographic and clinical variables: age, sex, race, BMI, smoking status, hypertension, and history of CVD) was 0.810 (95% CI: 0.794–0.825), with a sensitivity of 0.761 and a specificity of 0.715; meanwhile, the AUROC of the FLI (combined with demographic and clinical variables: age, sex, race, BMI, smoking status, hypertension, and history of CVD) was 0.838 (95% CI: 0.824–0.853), with a sensitivity of 0.824 and a specificity of 0.683 (Table 4 and Figure 2 and Figure 3).

## 4. Discussion

The present study has shown significant associations between the NPAR, the NLR, and the presence of NAFLD or advanced liver fibrosis. Per unit increases in both the NLR and NPAR were significantly associated with an increased risk of developing NAFLD, while neither the NLR nor the NPAR were significantly associated with higher odds of advanced fibrosis. The NPAR appears to be a satisfactory predictive biomarker for NAFLD.

The measurement and counting of peripheral blood leukocytes, especially neutrophils as in the NLR, is an inexpensive and widely used method to assess the presence of inflammation. Albumin is a medium-sized house-keeping protein with multiple functions, including osmoregulation, anti-oxidation, and anti-inflammation, accounting for more than half of the total human serum composition. Diseases, such as cirrhosis, are associated not only with reduced albumin synthesis but also specific alterations to its structure and function [25,26]. The NPAR, which combined the NLR and albumin, is used as a systemic inflammation-based predictor in patients with palliative pancreatic cancer [27], acute kidney injury [17], and septic shock [28]. The NPAR is also associated with mortality in patients with atrial fibrillation [29] and liver cirrhosis [30]. 

Chronic inflammation plays an important role in the development of NAFLD, which may range from simple steatosis to NASH, advanced liver fibrosis, cirrhosis, and finally to end-stage liver disease and hepatocellular carcinoma [31]. As such, NAFLD is a major cause of chronic, progressive liver injury, and being able to noninvasively diagnose and monitor progressive liver disease is essential. The NLR is an easily measurable inflammatory marker that has been utilized to prognosticate results in patients with cancer and coronary artery disease. Additionally, the NLR has demonstrated a correlation with the primary histological characteristics of NAFLD, specifically inflammation and fibrosis. The NLR or the lymphocyte count itself may be an important marker of immune function. A previous study examined the clinical utility of the mean platelet volume and NLR to predict the presence of fibrosis and NASH in patients with NAFLD [32]. Their study indicated that the mean platelet volume and NLR were elevated in patients with NASH versus non-NASH patients, as well as in advanced versus early fibrosis. An increased NLR was linked with an elevated risk of hepatitis B-associated hepatocellular carcinoma. A different study found that a heightened NLR and decreased lymphocyte counts were significantly associated with a greater risk of hepatocellular carcinoma in individuals with NAFLD [33]. Additionally, when the association between the NLR, the platelet-to-lymphocyte ratio, and NAFLD was explored, the relationship between the NLR and the platelet-to-lymphocyte ratio and NAFLD was non-linear after adjusting for potential confounders [34].

Fibrosis reflects the net result of fibrosis generation and fibrosis breakdown, which both occur simultaneously in progressive liver injury. Over time, however, fibrogenesis is likely to exceed the liver’s ability to degrade the accumulated extracellular matrix [12]. Currently, no study has yet to evaluate the association between the NPAR and NAFLD and advanced liver fibrosis. The present study found associations between the NPAR, the NLR, and the presence of NAFLD and advanced liver fibrosis. We observed that increases in both the NLR and NPAR were significantly associated with an increased risk of NAFLD. We also observed that neither the NLR nor the NPAR were significantly associated with the odds of advanced fibrosis. In the individuals without DM in this study, we found no significant association of the NPAR with NAFLD or advanced fibrosis. However, in the subjects with DM, an increased NPAR was significantly and independently associated with the odds of NAFLD.

A recent study explored the diagnostic value of the NLR as an indicator of steatosis and fibrosis severity [35]. That study reported that the NLR correlated significantly and positively with the degree of steatosis and fibrosis, and that an assessment by this method was highly sensitive and specific. A previous systematic review also explored the prognostic role of the NLR in the assessment of liver fibrosis and cirrhosis [15], finding that the NLR was significantly associated with the fibrosis stage and NAFLD activity score in patients with NAFLD, while for chronic hepatitis B patients, the NLR was inversely associated with the fibrosis stage. For patients with chronic hepatitis C, the NLR may not correlate significantly with the fibrosis stage. That review study also concluded that the NLR appeared to be particularly useful in predicting the prognosis of patients with cirrhosis. Therefore, the NLR may be associated with the liver fibrosis stage, especially in NAFLD patients.

This study found that the patients with NAFLD had significantly higher mean values of lymphocyte counts, neutrophil counts, and the NPAR, and lower mean serum albumin levels than those without NAFLD. Additionally, the patients with advanced fibrosis had a significantly higher mean NLR and NPAR than those without advanced fibrosis. These results suggest that a higher NPAR is significantly associated with the risk of NAFLD and may be a more effective biomarker for predicting NAFLD than albumin, the neutrophil percentage, and the NLR. Furthermore, the association between the NPAR and NAFLD was modified by DM status, which is a known risk factor for the development and progression of NAFLD [36]. In addition, DM itself also correlates with systemic inflammation [37]. Therefore, it is plausible that comorbid DM may mask the correlation between the NPAR and NAFLD.

## 5. Limitations 

The present study has several limitations. First, the retrospective design may limit the generalization of the results to other populations and does not allow ruling out the possibility of selection bias. Because timing is limited to individual interviews and examinations, long-term follow-up is also restricted. Moreover, the cross-sectional design does not allow causal inferences to be made. Although we included many covariates, unknown confounders may still exist. Because the data in this study were derived from the NHANES database, inaccurate reporting or recall bias may occur among participants when some parts of the NHANES survey are based on individual interviews and self-reported questionnaires. Further prospective studies of the relationships between the NPAR, the NLR, and NAFLD are warranted to confirm the results of the present study, particularly the predictive value of the NPAR in NAFLD.

## 6. Conclusions

Significant associations are shown between the NPAR, the NLR, and the presence of NAFLD or advanced liver fibrosis in the general population, but not in patients with DM. The novel biomarker NPAR, combined with participants’ medical history and clinical characteristics, appears to be a valid predictive factor for NAFLD in a nondiabetic population. Evidence of the potential predictive and diagnostic utility of the NPAR as a biomarker for NAFLD may help clinicians to refine the diagnosis and treatment of chronic liver disease.

## Figures and Tables

**Figure 1 nutrients-15-01892-f001:**
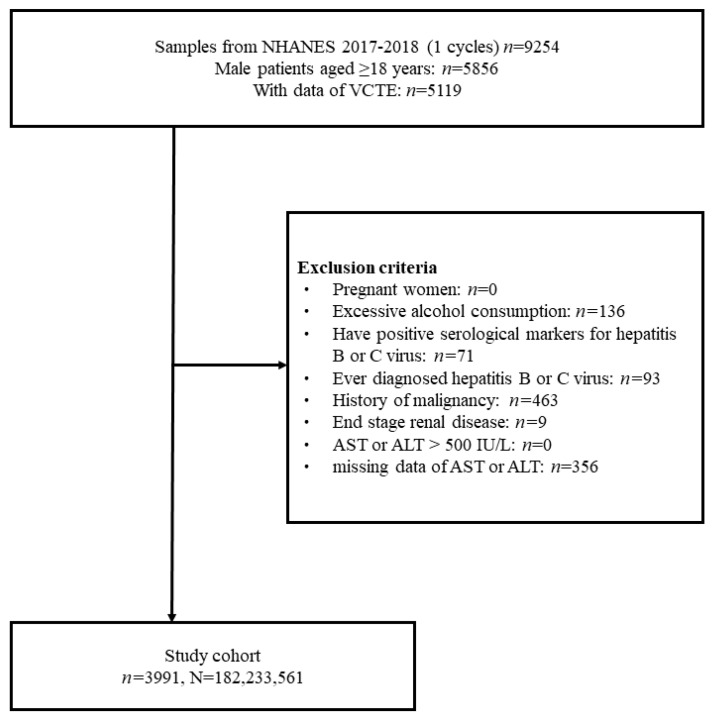
Flow diagram of study cohort selection.

**Figure 2 nutrients-15-01892-f002:**
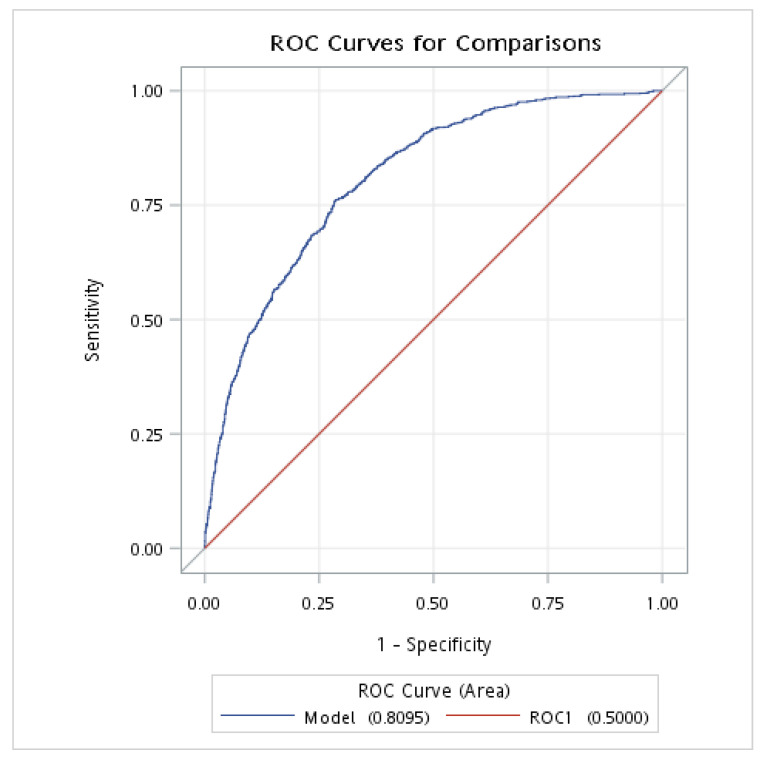
The AUROC of NPAR for NAFLD in individuals without DM. AUROC, area under the receiver operating characteristic curve; DM, diabetes mellitus; NAFLD, nonalcoholic fatty liver disease; NPAR, neutrophil-percentage-to-albumin ratio. This analysis was adjusted for age (continuous), gender, race, BMI, smoking, hypertension, and history of CVD.

**Figure 3 nutrients-15-01892-f003:**
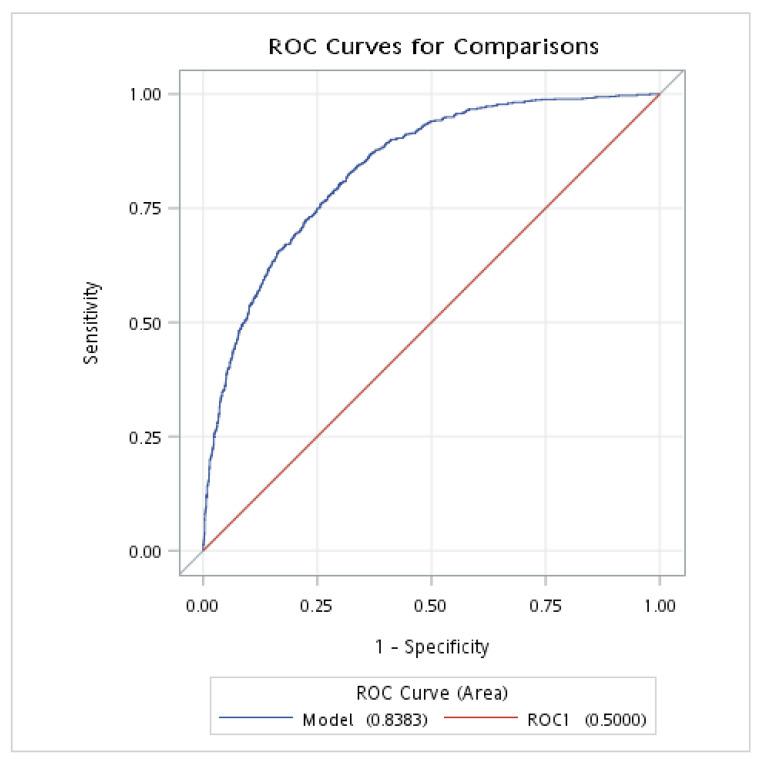
The AUROC of FLI for NAFLD in individuals without DM. AUROC, area under the receiver operating characteristic curve; DM, diabetes mellitus; NAFLD, nonalcoholic fatty liver disease; FLI, fatty liver index. This analysis was adjusted for age (continuous), gender, race, BMI, smoking, hypertension, and history of CVD.

**Table 1 nutrients-15-01892-t001:** Characteristics of the study cohort.

	Total	NAFLD ^a^			Advanced Liver Fibrosis		
Study Variables		Yes	No		Yes	No	
	*n* = 3991	*n* = 1355	*n* = 2474	*p*-Value	*n* = 162	*n* = 3829	*p*-Value
NLR	2.1 ± 0.03	2.1 ± 0.04	2.1 ± 0.04	0.595	2.2 ± 0.07	2.1 ± 0.03	0.068
Q1	1004 (21.6)	332 (19.8)	649 (23.1)	0.129	23 (11.4)	981 (22.0)	0.128
Q2	989 (26.5)	325 (24.2)	621 (27.1)		43 (34.4)	946 (26.1)	
Q3	1004 (25.4)	339 (27.1)	623 (24.6)		42 (24.5)	962 (25.4)	
Q4	992 (26.6)	359 (28.8)	579 (25.2)		54 (29.7)	938 (26.4)	
Missing	2	0	2		0	2	
NPAR	14.1 ± 0.1	14.3 ± 0.1	13.9 ± 0.1	**0.013**	14.9 ± 0.3	14.0 ± 0.1	**0.013**
Q1	1009 (24.4)	324 (22.0)	661 (26.3)	**0.009**	24 (13.0)	985 (24.8)	0.081
Q2	999 (25.4)	321 (22.9)	649 (26.8)		29 (23.8)	970 (25.5)	
Q3	989 (24.5)	354 (27.7)	594 (22.7)		41 (28.0)	948 (24.4)	
Q4	992 (25.7)	356 (27.4)	568 (24.2)		68 (35.3)	924 (25.3)	
Missing	2	0	2		0	2	
FLI	54.5 ± 1.5	77.7 ± 1.6	40.4 ± 1.5	**<0.001**	92.1 ± 1.6	53.1 ± 1.5	**<0.001**
Q1	963 (26.1)	41 (2.6)	920 (39.7)	**<0.001**	2 (0.7)	961 (27.0)	**<0.001**
Q2	962 (23.7)	239 (17.6)	718 (28.0)		5 (2.3)	957 (24.5)	
Q3	961 (24.5)	419 (29.4)	512 (22.5)		30 (16.2)	931 (24.8)	
Q4	964 (25.7)	623 (50.3)	235 (9.9)		106 (80.9)	858 (23.7)	
Missing	122	33	89		19	122	
Age, years	45.3 ± 0.7	48.7 ± 0.7	43.2 ± 0.7	**<0.001**	50.7 ± 1.7	45.1 ± 0.7	**0.005**
18–49	2074 (58.7)	590 (50.1)	1432 (63.9)	**<0.001**	52 (45.1)	2022 (59.2)	0.100
50–59	642 (18.2)	273 (22.5)	337 (15.6)		32 (24.9)	610 (17.9)	
60–69	744 (13.6)	304 (16.3)	390 (11.9)		50 (19.1)	694 (13.4)	
70–79	351 (6.9)	137 (8.9)	194 (5.8)		20 (7.5)	331 (6.9)	
80+	180 (2.6)	51 (2.2)	121 (2.7)		8 (3.5)	172 (2.5)	
Sex							
Male	1922 (48.8)	749 (56.0)	1078 (44.6)	**<0.001**	95 (57.1)	1827 (48.5)	0.200
Female	2069 (51.2)	606 (44.0)	1396 (55.4)		67 (42.9)	2002 (51.5)	
Race							
Non-Hispanic White	1285 (60.4)	450 (60.1)	770 (60.2)	0.090	65 (65.7)	1220 (60.2)	0.267
Non-Hispanic Black	913 (11.2)	259 (8.9)	628 (12.6)		26 (8.0)	887 (11.4)	
Hispanic	399 (7.4)	125 (6.6)	258 (8.0)		16 (4.2)	383 (7.5)	
Others	1394 (21.0)	521 (24.5)	818 (19.2)		55 (22.1)	1339 (21.0)	
BMI, kg/m^2^ (Missing = 27)	29.8 ± 0.3	33.6 ± 0.5	27.1 ± 0.3	**<0.001**	41.9 ± 1.1	29.3 ± 0.3	**<0.001**
Poverty-to-income ratio							
Not poor (>1)	2822 (86.8)	983 (88.2)	1718 (85.9)	0.706	121 (89.5)	2701 (86.7)	0.493
Poor (≤1)	665 (13.2)	201 (11.8)	442 (14.1)		22 (10.5)	643 (13.3)	
Missing	485	171	314		39	465	
Education level							
High school and above	3187 (89.2)	1085 (89.6)	1978 (89.2)	0.515	124 (87.0)	3063 (89.3)	0.420
Never attended high school	749 (10.8)	262 (10.4)	450 (10.8)		37 (13.0)	712 (10.7)	
Missing	54	8	46		1	54	
Smoking status							
Never	2470 (60.9)	784 (57.3)	1598 (63.2)	**<0.001**	88 (53.6)	2382 (61.2)	**0.036**
Former	871 (23.5)	360 (27.7)	457 (20.8)		54 (34.2)	817 (23.1)	
Current smoker	650 (15.5)	211 (15.0)	419 (15.9)		20 (12.2)	630 (15.6)	
DM	776 (13.5)	399 (23.3)	291 (6.6)	**<0.001**	86 (47.0)	690 (12.2)	**<0.001**
Hypertension	1593 (33.7)	710 (48.2)	773 (24.6)	**<0.001**	110 (64.5)	1483 (32.6)	**<0.001**
History of CVD	361 (7.1)	142 (9.1)	190 (5.4)	**0.002**	29 (18.2)	332 (6.7)	**<0.001**
CKD	289 (6.6)	103 (7.7)	166 (6.0)	0.076	20 (8.0)	269 (6.5)	0.509
Laboratory data							
AST, U/L	21.6 ± 0.2	22.4 ± 0.3	20.7 ± 0.3	**0.001**	29.5 ± 2.5	21.3 ± 0.2	**0.005**
ALT, U/L	22.7 ± 0.4	27.6 ± 0.8	19.6 ± 0.3	**<0.001**	33.3 ± 2.3	22.3 ± 0.4	**<0.001**
Total cholesterol, mg/dL (Missing = 4)	187.7 ± 1.7	192.0 ± 2.1	185.5 ± 1.6	**0.002**	187.8 ± 5.8	187.7 ± 1.6	0.982
Triglycerides, mg/dL	141.0 ± 3.2	182.8 ± 4.6	117.4 ± 2.1	**<0.001**	180.2 ± 15.2	139.5 ± 3.2	**0.017**
Platelet, 10^9^/L(Missing = 1)	246.9 ± 2.7	253.2 ± 3.2	244.4 ± 2.9	**0.002**	236.0 ± 5.8	247.3 ± 2.7	0.056
Lymphocyte count, 10^9^/L (Missing = 2)	2.2 ± 0.03	2.4 ± 0.04	2.1 ± 0.02	**<0.001**	2.3 ± 0.1	2.2 ± 0.03	**0.083**
Neutrophil count, 10^9^/L (Missing = 2)	4.3 ± 0.1	4.6 ± 0.1	4.1 ± 0.1	**<0.001**	4.7 ± 0.1	4.3 ± 0.1	**0.001**
Albumin, g/dL	41.1 ± 0.2	40.8 ± 0.2	41.3 ± 0.2	**0.01** **0**	39.7 ± 0.3	41.1 ± 0.2	**<0.001**
HbA1c, % (Missing = 1)	5.7 ± 0.02	6.0 ± 0.04	5.4 ± 0.02	**<0.001**	6.4 ± 0.13	5.6 ± 0.02	**<0.001**

^a^ Exclude advanced fibrosis. AST, aspartate aminotransferase; ALT, alanine aminotransferase; NLR, neutrophil-to-lymphocyte ratio; NPAR, neutrophil-percentage-to-albumin ratio; FLI, fatty liver index; NAFLD, nonalcoholic fatty liver disease; DM, diabetes mellitus; CKD, chronic kidney disease; CVD, cardiovascular disease. Continuous variables are presented as mean ± SE; categorical variables are presented as unweighted counts (weighted percentage). *p*-values < 0.05 are shown in bold.

**Table 2 nutrients-15-01892-t002:** Multivariable regression analysis of associations between NLR, NPAR, FLI, and presence of NAFLD and advanced liver fibrosis.

	NAFLD		Advanced Fibrosis	
	aOR ^a^ (95% CI)	*p*-Value	aOR ^a^ (95% CI)	*p*-Value
NLR	**0.88 (0.77–1.00)**	**0.032**	0.79 (0.60–1.04)	0.068
Q1	1		1	
Q2	0.86 (0.58–1.28)	0.416	1.91 (0.65–5.66)	0.202
Q3	0.96 (0.72–1.29)	0.784	0.87 (0.29–2.56)	0.776
Q4	0.73 (0.48–1.10)	0.101	0.69 (0.24–1.96)	0.453
NPAR	**0.93 (0.88–0.99)**	**0.017**	**0.89 (0.79–0.99)**	**0.019**
Q1	1		1	
Q2	0.86 (0.59–1.27)	0.420	1.43 (0.50–4.07)	0.469
Q3	0.98 (0.68–1.42)	0.923	1.12 (0.47–2.64)	0.781
Q4	**0.64 (0.43–0.95)**	**0.01** **6**	0.63 (0.24–1.68)	0.317
FLI	**1.04 (1.03–1.05)**	**<0.001**	**1.03 (1.00–1.05)**	**0.022**
Q1	1		1	
Q2	6.52 (3.36–12.64)	**<0.001**	1.17 (0.17–8.21)	0.860
Q3	**11.07 (5.27–23.26)**	**<0.001**	4.46 (0.82–24.19)	0.060
Q4	**33.14 (14.35–76.51)**	**<0.001**	5.41 (0.77–38.05)	0.065

aOR, adjusted odds ratio; NLR, neutrophil-to-lymphocyte ratio; NPAR, neutrophil-percentage-to-albumin ratio; FLI, fatty liver index. *p*-values < 0.05 are shown in bold. ^a^ Adjusted for age (continuous), sex, race, BMI, smoking, DM, hypertension, and history of CVD.

**Table 3 nutrients-15-01892-t003:** Associations between NPAR, FLI, and presence of NAFLD and advanced fibrosis stratified by DM status.

	NAFLD		Advance Fibrosis	
	aOR ^a^ (95% CI)	*p*-Value	aOR ^a^ (95% CI)	*p*-Value
Individuals with DM				
NPAR, continuous	1.02 (0.90–1.16)	0.733	0.89 (0.73–1.08)	0.200
NPAR, in quartiles				
Q1	1		1	
Q2	1.38 (0.57–3.31)	0.437	0.86 (0.21–3.59)	0.827
Q3	1.14 (0.41–3.17)	0.779	0.55 (0.13–2.28)	0.368
Q4	1.13 (0.39–3.22)	0.811	0.41 (0.13–1.24)	0.085
FLI, continuous	**1.04 (1.02–1.06)**	**<0.001**	1.03 (0.99–1.07)	0.112
FLI, in quartiles				
Q1	1		1	
Q2	**3.29** **(1.02–10.60)**	**0.03** **0**	N/A	N/A
Q3	**6.01 (1.48–24.44)**	**0.006**	N/A	N/A
Q4	**9.23 (1.39–61.30)**	**0.012**	N/A	N/A
Individuals without DM				
NPAR, continuous	**0.91 (0.85–0.97)**	**0.002**	0.88 (0.75–1.02)	0.073
NPAR, in quartiles				
Q1	1		1	
Q2	0.81 (0.54–1.22)	0.267	2.09 (0.61–7.14)	0.199
Q3	0.95 (0.68–1.33)	0.748	1.78 (0.47–6.72)	0.357
Q4	**0.56 (0.36–0.88)**	**0.006**	0.87 (0.25–2.98)	0.808
FLI, continuous	**1.04 (1.03–1.05)**	**<0.001**	1.02 (0.99–1.05)	0.105
FLI, in quartiles				
Q1	1		1	
Q2	**7.00 (3.43–14.29)**	**<0.001**	0.30 (0.03–3.18)	0.275
Q3	**12.09 (5.48–26.65)**	**<0.001**	2.82 (0.46–17.43)	0.226
Q4	**41.01 (16.30–103.17)**	**<0.001**	2.36 (0.25–22.56)	0.418

aOR, adjusted odds ratio; DM, diabetes mellitus; NAFLD, nonalcoholic fatty liver disease; NPAR, neutrophil-percentage-to-albumin ratio; FLI, fatty liver index; N/A, not applicable; Q, quartile. *p*-values < 0.05 are shown in bold. ^a^ Adjusted for age (in years), sex, race, BMI, smoking, hypertension, and history of CVD.

**Table 4 nutrients-15-01892-t004:** ROC analysis of the diagnostic performances of NPAR for NAFLD and FLI in individuals without DM.

Variable	AUROC ^a^	95% CI	Youden Index	Sensitivity	Specificity
NPAR	0.810	0.794–0.825	0.476	0.761	0.715
FLI	0.838	0.824–0.853	0.507	0.824	0.683

DM, diabetes mellitus; NAFLD, nonalcoholic fatty liver disease; NPAR, neutrophil-percentage-to-albumin ratio; FLI, fatty liver index; AUROC, area under the receiver operating characteristic curve; CI, confidence interval. ^a^ Adjusted for age (in years), sex, race, BMI, smoking, hypertension, and history of CVD.

## Data Availability

The data presented in this study are available on request from the corresponding author. The data are not publicly available due to restrictions concerning privacy and ethical reasons.

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
