# Peer review of "Predictive Role of Neutrophil-Percentage-to-Albumin Ratio (NPAR) in Nonalcoholic Fatty Liver Disease and Advanced Liver Fibrosis in Nondiabetic US Adults: Evidence from NHANES 2017–2018"

_nutrients, 2023, doi:10.3390/nu15081892_

Round 1
Reviewer 1 Report
It is cross-sectional study searching for a predictive role of neutrophil-percentage-to albumin ratio (NPAR) in diagnosis of non-alcoholic fatty liver disease (NAFLD) and advanced fibrosis in non-diabetic adults in NHANES population (cycle 2017-2018).
This study is continuation of many other studies using easily available hematological parameters to assess severity of organ damage or prognosis in metabolic, neoplastic or inflammatory diseases. Therefore, this kind of biomarkers shows very low specificity to the liver, in general and to NAFLD, in particular.
This study has no exploring aims, but has purely practical targets, therefore, should only be reviewed from this perspective.
I have several concerns about this study.
1. The total number of analyzed subjects was 3991. Looking at numbers given in Table 1 it is clear that 369 patients with “advanced fibrosis” were included to NAFLD group. On one hand it could be argued that the term of NAFLD is an umbrella covering whole histological spectrum of NAFLD, however, on the other hand for the purpose of this study it would more reasonable to exclude persons with progressive disease from the remaining NAFLD persons who are deemed to belong to non-NASH or at least without significant fibrosis population.
2. Following principles from therapeutic trials also the novel diagnostic biomarkers should be compared to already accepted indices to detect liver steatosis (e.g. FLI, HSI) to evidence their non-inferiority or superiority.
3. VCTE cut-off for diagnosis of advanced fibrosis was 8,8 kPa in this study. However, in majority of recommendations 12 kPa is used for stratifying patients into low- and high-risk populations, and the interval between 8-12 kPa is interpreted as indeterminate zone, requiring use of other examinations (e.g., ELF test, MRE or biopsy). It is interesting to see what result were, if 12 kPa was taken as cut-off value, and how far this alteration would limit the size of population with progressive fibrosis.
4. Multivariate analysis showed that NPAR was significantly associated with increased risk of developing NAFLD, but this biomarker was not predictive for advanced fibrosis. Mean values of NPAR and also lymphocyte and neutrophil counts, AST, ALT, total cholesterol, triglycerides, and HbA1c were significantly higher in participants with NAFLD than in those without NAFLD or advanced liver fibrosis. This suggests that in terms of metabolic and hepatic inflammatory biomarkers individuals without NAFLD resembled those with progressive NAFLD-related fibrosis. It is not good for clinical test utility. This finding may reflect the well-known process of metabolic syndrome withdrawal in late stage of NAFLD, characterized by decline of aminotransferases, lipids and glucose plasma levels. Moreover, there was inverse relationship in behavior of two components of NPAR at the stage of advanced fibrosis, as albumin showed decreasing and neutrophil count increasing tendency. This relationship does not favor NPAR as important test, as the present priority is looking for a test detecting progressive disease and not any steatosis, irrespective of its etiology.
Minor remark
The phrase “Subjects without NAFLD or progressive fibrosis had significantly higher mean serum albumin levels than those with NAFLD or progressive fibrosis” is not clear as progressive fibrosis is mentioned in both self-excluding messages.
It is cross-sectional study searching for a predictive role of neutrophil-percentage-to albumin ratio (NPAR) in diagnosis of non-alcoholic fatty liver disease (NAFLD) and advanced fibrosis in non-diabetic adults in NHANES population (cycle 2017-2018).
This study is continuation of many other studies using easily available hematological parameters to assess severity of organ damage or prognosis in metabolic, neoplastic or inflammatory diseases. Therefore, this kind of biomarkers shows very low specificity to the liver, in general and to NAFLD, in particular.
This study has no exploring aims, but has purely practical targets, therefore, should only be reviewed from this perspective.
I have several concerns about this study.
1. The total number of analyzed subjects was 3991. Looking at numbers given in Table 1 it is clear that 369 patients with “advanced fibrosis” were included to NAFLD group. On one hand it could be argued that the term of NAFLD is an umbrella covering whole histological spectrum of NAFLD, however, on the other hand for the purpose of this study it would more reasonable to exclude persons with progressive disease from the remaining NAFLD persons who are deemed to belong to non-NASH or at least without significant fibrosis population.
2. Following principles from therapeutic trials also the novel diagnostic biomarkers should be compared to already accepted indices to detect liver steatosis (e.g. FLI, HSI) to evidence their non-inferiority or superiority.
3. VCTE cut-off for diagnosis of advanced fibrosis was 8,8 kPa in this study. However, in majority of recommendations 12 kPa is used for stratifying patients into low- and high-risk populations, and the interval between 8-12 kPa is interpreted as indeterminate zone, requiring use of other examinations (e.g., ELF test, MRE or biopsy). It is interesting to see what result were, if 12 kPa was taken as cut-off value, and how far this alteration would limit the size of population with progressive fibrosis.
4. Multivariate analysis showed that NPAR was significantly associated with increased risk of developing NAFLD, but this biomarker was not predictive for advanced fibrosis. Mean values of NPAR and also lymphocyte and neutrophil counts, AST, ALT, total cholesterol, triglycerides, and HbA1c were significantly higher in participants with NAFLD than in those without NAFLD or advanced liver fibrosis. This suggests that in terms of metabolic and hepatic inflammatory biomarkers individuals without NAFLD resembled those with progressive NAFLD-related fibrosis. It is not good for clinical test utility. This finding may reflect the well-known process of metabolic syndrome withdrawal in late stage of NAFLD, characterized by decline of aminotransferases, lipids and glucose plasma levels. Moreover, there was inverse relationship in behavior of two components of NPAR at the stage of advanced fibrosis, as albumin showed decreasing and neutrophil count increasing tendency. This relationship does not favor NPAR as important test, as the present priority is looking for a test detecting progressive disease and not any steatosis, irrespective of its etiology.
Minor remark
The phrase “Subjects without NAFLD or progressive fibrosis had significantly higher mean serum albumin levels than those with NAFLD or progressive fibrosis” is not clear as progressive fibrosis is mentioned in both self-excluding messages.
Author Response
Reviewer 1:
It is cross-sectional study searching for a predictive role of neutrophil-percentage-to albumin ratio (NPAR) in diagnosis of non-alcoholic fatty liver disease (NAFLD) and advanced fibrosis in non-diabetic adults in NHANES population (cycle 2017-2018).
This study is continuation of many other studies using easily available hematological parameters to assess severity of organ damage or prognosis in metabolic, neoplastic or inflammatory diseases. Therefore, this kind of biomarkers shows very low specificity to the liver, in general and to NAFLD, in particular.
This study has no exploring aims, but has purely practical targets, therefore, should only be reviewed from this perspective.
Author Response:
Thank you very much for your comment.
I have several concerns about this study.
- The total number of analyzed subjects was 3991. Looking at numbers given in Table 1 it is clear that 369 patients with “advanced fibrosis” were included to NAFLD group. On one hand it could be argued that the term of NAFLD is an umbrella covering whole histological spectrum of NAFLD, however, on the other hand for the purpose of this study it would more reasonable to exclude persons with progressive disease from the remaining NAFLD persons who are deemed to belong to non-NASH or at least without significant fibrosis population.
Author Response:
We increase the cut-off value of advanced liver fibrosis to VCTE ≥ 12 kPa and conducted analysis again.
- Following principles from therapeutic trials also the novel diagnostic biomarkers should be compared to already accepted indices to detect liver steatosis (e.g. FLI, HSI) to evidence their non-inferiority or superiority.
Author Response:
The FLI is compared in Table 2.
- VCTE cut-off for diagnosis of advanced fibrosis was 8,8 kPa in this study. However, in majority of recommendations 12 kPa is used for stratifying patients into low- and high-risk populations, and the interval between 8-12 kPa is interpreted as indeterminate zone, requiring use of other examinations (e.g., ELF test, MRE or biopsy). It is interesting to see what result were, if 12 kPa was taken as cut-off value, and how far this alteration would limit the size of population with progressive fibrosis.
Author Response:
We increase the cut-off value of advanced liver fibrosis to VCTE ≥ 12 kPa and conducted analysis again.
- Multivariate analysis showed that NPAR was significantly associated with increased risk of developing NAFLD, but this biomarker was not predictive for advanced fibrosis. Mean values of NPAR and also lymphocyte and neutrophil counts, AST, ALT, total cholesterol, triglycerides, and HbA1c were significantly higher in participants with NAFLD than in those without NAFLD or advanced liver fibrosis. This suggests that in terms of metabolic and hepatic inflammatory biomarkers individuals without NAFLD resembled those with progressive NAFLD-related fibrosis. It is not good for clinical test utility. This finding may reflect the well-known process of metabolic syndrome withdrawal in late stage of NAFLD, characterized by decline of aminotransferases, lipids and glucose plasma levels. Moreover, there was inverse relationship in behavior of two components of NPAR at the stage of advanced fibrosis, as albumin showed decreasing and neutrophil count increasing tendency. This relationship does not favor NPAR as important test, as the present priority is looking for a test detecting progressive disease and not any steatosis, irrespective of its etiology.
Author Response:
The revised data show that NPAR and FLI (fatty liver index) are both significant associated with NAFLD and advanced fibrosis in subjects without diabetes.
Minor remark
The phrase “Subjects without NAFLD or progressive fibrosis had significantly higher mean serum albumin levels than those with NAFLD or progressive fibrosis” is not clear as progressive fibrosis is mentioned in both self-excluding messages.
Author Response:
The sentence is modified to “The mean blood albumin levels of subjects without NAFLD or advancing fibrosis were considerably greater than those of individuals with these conditions”.
Reviewer 2 Report
Nonalcoholic fatty liver disease ( NAFLD) is the most commom cause of liver dieseases in the Western world today. The surge of NAFLD is closely linked to pandemic multisystem metabolic diseases such as diabetes mellitus and obesity. Factors such as insulin resistance, abdominal obesity, genetic and environmental factors are closely associated with the progression of this disease from bland hepatic steatosis to nonalcoholic fatty liver hepatitis ( NASH). NASH has a multi-factorial genesis which, alongside genetic and lyfestile factors ( malnutrition anf hyper-nutrition) that lead to excessive fat accumulation in the the liver, endotoxins, and proinflammatory cytokines contribute to chronic inflammation of the liver. NASH generally considered a risk factor for the development of cirrhosis or hepatocellular carcinoma. Early diagnosis and assessment of the severity of fibrosis are of significant prognostic importance for monitoring the progression of fibrosis and treatment, especially of concomitant diseases, such as diabetes mellitus and obesity. The biomarkers NPAR ( neutrophil-percentage-to-albumin ratio) and NLR ( neutrophil-to-lymphocyte ratio) presented by the authors are already known as prognostic markers in tumour and cardiovascular diseases. Based on the data from a NHANES ( National Health and Nutrition Examination Survey) study, a clinical examination, specialised measurements, and lab tests were performed and evaluated retrospectively in addition to a"household interview". According to the findings presented, these markers may also be interest with respect to NAFLD, such as PNAR as a predictive biomarker for NAFLD patients without diabetes mellitus and NLR as a marker for assessing the severity of fibrosis. The latter in particular would be an important addition to the gold standard of liver biopsy with histological evaluation and FibroScan.
However, the findings presented absolutely require confirmation in prospective studies before widespread clinical application.
What is confusing and therefore unclear to the reviewer is the fact that, from available data, it is not evident how many patients were included in the studies and therefore al further statements cannot be clearly assessed. For example, in 3.1 Results , " 9,254 interviewed patients" are initially cited, later this figure is 5,119, and after excluding additional patients with insufficient data, a population of " 182,233, 561 US adults" is ultimately cited. These numbers are also reflected in Fig. 1.
Due to the inconsistencies regarding the number paients included in the study, it is of course impossible to evaluate the data based on them ( statistics, ect. )
Due to the overall highly relevant and interesting subject matter, I recommend a general revision of the manuscript.
Author Response
Reviewer 2
Nonalcoholic fatty liver disease ( NAFLD) is the most commom cause of liver dieseases in the Western world today. The surge of NAFLD is closely linked to pandemic multisystem metabolic diseases such as diabetes mellitus and obesity. Factors such as insulin resistance, abdominal obesity, genetic and environmental factors are closely associated with the progression of this disease from bland hepatic steatosis to nonalcoholic fatty liver hepatitis ( NASH). NASH has a multi-factorial genesis which, alongside genetic and lyfestile factors ( malnutrition anf hyper-nutrition) that lead to excessive fat accumulation in the the liver, endotoxins, and proinflammatory cytokines contribute to chronic inflammation of the liver. NASH generally considered a risk factor for the development of cirrhosis or hepatocellular carcinoma. Early diagnosis and assessment of the severity of fibrosis are of significant prognostic importance for monitoring the progression of fibrosis and treatment, especially of concomitant diseases, such as diabetes mellitus and obesity. The biomarkers NPAR ( neutrophil-percentage-to-albumin ratio) and NLR ( neutrophil-to-lymphocyte ratio) presented by the authors are already known as prognostic markers in tumour and cardiovascular diseases. Based on the data from a NHANES ( National Health and Nutrition Examination Survey) study, a clinical examination, specialised measurements, and lab tests were performed and evaluated retrospectively in addition to a"household interview". According to the findings presented, these markers may also be interest with respect to NAFLD, such as PNAR as a predictive biomarker for NAFLD patients without diabetes mellitus and NLR as a marker for assessing the severity of fibrosis. The latter in particular would be an important addition to the gold standard of liver biopsy with histological evaluation and FibroScan.
However, the findings presented absolutely require confirmation in prospective studies before widespread clinical application.
What is confusing and therefore unclear to the reviewer is the fact that, from available data, it is not evident how many patients were included in the studies and therefore al further statements cannot be clearly assessed. For example, in 3.1 Results , " 9,254 interviewed patients" are initially cited, later this figure is 5,119, and after excluding additional patients with insufficient data, a population of " 182,233, 561 US adults" is ultimately cited. These numbers are also reflected in Fig. 1.
Author Response:
We revised section 3.1 to explain the procedure of cohort selection. The “N” is represented because readers often want to know how many persons in the whole US population the analysis can represent as this study analyzes NHANES data. The "N" value is thus only used for this specific reason.
Due to the inconsistencies regarding the number paients included in the study, it is of course impossible to evaluate the data based on them ( statistics, ect. )
Author Response:
The final cohort contains 3,991 subjects for statistical analysis.
Due to the overall highly relevant and interesting subject matter, I recommend a general revision of the manuscript.
Reviewer 3 Report
This study aimed to evaluate the association between a compound variable NPAR derived from hematologic parameters in patients with NAFLD or advanced liver fibrosis based on a public database NHANES. The analysis process is clearly described, and the findings are of clinical significance. However, I have the following concerns for the authors’ consideration:
Abstract
* Line 6: the second NPAR seems redundant.
* Is “progressive fibrosis” the same thing with “advanced fibrosis”. Please unify the terminology.
* “NHANES participants with and without liver stiffness indicated by VCTE and controlled attenuation parameter (CAP) were enrolled” – Please clarify this sentence.
* There is not description about predictive performance of NPAR in this part but the conclusion claimed “the novel biomarker NPAR demonstrates good predictive value for NAFLD”.
Introduction
*The second paragraph talked about the outcomes of NAFLD, but no outcomes evaluated in this study. The relevance of this paragraph with the study aims is weak.
*The third paragraph is talking about “early diagnosis of fibrosis”, but this study actually evaluated the discriminative ability of NPAR for NAFLD – a little off-target?
M&M
1. “Ethical statement” is partly overlapping with the second paragraph in “study design and data source”. Please confirm it.
2. Please provide support reference for the statement of CAP accuracy “(sensitivity: 93.7%, specificity: 91.1%)”.
Results
*I feel confused about this statement “…who had data of VCTE and CAP were eligible for inclusion. After excluding 356 subjects with missing information on VCTE, CAP…” – why patients with VCTE and CAP data can still missed VCTE and CAP information. Please make it clear.
*please briefly describe how the number 182,233,561 was calculated or provide the related link describing the calculation method.
* “When regarded as quartiles, compared with the lowest quartile (Q1), subjects with the highest quartile of NPAR (Q4) were significantly more likely to have NAFLD (aOR=1.08; 95% CI: 1.01- 1.14)”- should the aOR be 1.57? Please double check it.
*The description about the Tables 3 seems confusing. Please double check “with” or “without DM”. Also check the other parts.
Discussion
*Only 7 references cited in this paper, seems not a comprehensive discussion here.
*Some sentences have appeared in the “introduction” part, please provide a discussion based on what your findings and compared with peers’ findings. Do not simply repeat the description in the introduction.
Conclusions
*The findings of this study did not show a significant association between NPAR, NLR and advanced fibrosis, why “Significant associations are shown between NPAR, NLR, and the presence of NAFLD or advanced liver fibrosis” was still stated? Confused.
Figures
Figure 2. Please state it is adjusted AUROC in the legend.
Tables
Table 2. It seems odd that the overall result was significant for NLR in NALFD (p = 0.015), but no results significant in Q2,Q3 and Q4. Please double check the results.
Author Response
Reviewer 3
This study aimed to evaluate the association between a compound variable NPAR derived from hematologic parameters in patients with NAFLD or advanced liver fibrosis based on a public database NHANES. The analysis process is clearly described, and the findings are of clinical significance. However, I have the following concerns for the authors’ consideration:
Abstract
* Line 6: the second NPAR seems redundant.
Author Response:
The second NPAR is deleted.
* Is “progressive fibrosis” the same thing with “advanced fibrosis”. Please unify the terminology.
Author Response:
The term “progressive fibrosis” is changed to “advanced fibrosis”.
* “NHANES participants with and without liver stiffness indicated by VCTE and controlled attenuation parameter (CAP) were enrolled” – Please clarify this sentence.
Author Response:
The sentence is modified to “NHANES participants with complete information of vibration-controlled transient elastography (VCTE) and controlled attenuation parameter (CAP) were enrolled.”.
* There is not description about predictive performance of NPAR in this part but the conclusion claimed “the novel biomarker NPAR demonstrates good predictive value for NAFLD”.
Author Response:
The last two sentences are modified to avoid the predictive description.
Introduction
*The second paragraph talked about the outcomes of NAFLD, but no outcomes evaluated in this study. The relevance of this paragraph with the study aims is weak.
Author Response:
The second paragraph is used to stress the health impact of NAFLD and fibrosis, and lead to the importance of early detection of NAFLD in the third paragraph. Although the relevance of this paragraph with the aim of this study is weak, it tells a full story when combined with paragraph 1 and 3.
*The third paragraph is talking about “early diagnosis of fibrosis”, but this study actually evaluated the discriminative ability of NPAR for NAFLD – a little off-target?
Author Response:
The discriminative ability of NPAR for both NAFLD and advanced liver fibrosis is studied in this study. NAFLD is added in the paragraph along with liver fibrosis.
M&M
- “Ethical statement” is partly overlapping with the second paragraph in “study design and data source”. Please confirm it.
Author Response:
The second paragraph of study design and data source is incorporated to “ethical statement”.
- Please provide support reference for the statement of CAP accuracy “(sensitivity: 93.7%, specificity: 91.1%)”.
Author Response:
We add the reference which show CAP can be used to accurately predict NAFLD and liver fibrosis.
Results
*I feel confused about this statement “…who had data of VCTE and CAP were eligible for inclusion. After excluding 356 subjects with missing information on VCTE, CAP…” – why patients with VCTE and CAP data can still missed VCTE and CAP information. Please make it clear.
Author Response:
The section 3.1 is revised.
*please briefly describe how the number 182,233,561 was calculated or provide the related link describing the calculation method.
Author Response:
The sentence “We used weighted samples, stratum, and cluster of the NHANES database and the SUR-VEY procedure in SAS performs to calculate the weighted population of this study.” is added in the statistical analysis section.
* “When regarded as quartiles, compared with the lowest quartile (Q1), subjects with the highest quartile of NPAR (Q4) were significantly more likely to have NAFLD (aOR=1.08; 95% CI: 1.01- 1.14)”- should the aOR be 1.57? Please double check it.
Author Response:
We checked and revised the description about the Tables 2.
*The description about the Tables 3 seems confusing. Please double check “with” or “without DM”. Also check the other parts.
Author Response:
We checked and revised the description about the Tables 3.
Discussion
*Only 7 references cited in this paper, seems not a comprehensive discussion here.
Author Response:
The discussion section is revised with new references.
*Some sentences have appeared in the “introduction” part, please provide a discussion based on what your findings and compared with peers’ findings. Do not simply repeat the description in the introduction.
Author Response:
The discussion section is revised.
Conclusions
*The findings of this study did not show a significant association between NPAR, NLR and advanced fibrosis, why “Significant associations are shown between NPAR, NLR, and the presence of NAFLD or advanced liver fibrosis” was still stated? Confused.
Author Response:
Although the association is not significant in patients with DM, NPAR and NLR are associated with the presence of NAFLD or advanced liver fibrosis in general population as shown in Table 2. To avoid the confusion, the sentence is modified to “Significant associations are shown between NPAR, NLR, and the presence of NAFLD or advanced liver fibrosis in general population, but not in patients with DM.”.
Figures
Figure 2. Please state it is adjusted AUROC in the legend.
Author Response:
We added the description about "adjusted AUROC” underneath Figure 2.
Tables
Table 2. It seems odd that the overall result was significant for NLR in NALFD (p = 0.015), but no results significant in Q2, Q3 and Q4. Please double check the results.
Author Response:
The outcomes are accurate after a second check. Because the quartile was not the best cut group, there may not be any statistically significant results in the categorical variable of NLR.

Round 2
Reviewer 1 Report
Thank you for replies and revisions
Reviewer 2 Report
After consistent revision, the authors now present an easily readable and comprehensible work, despite the large number of figures in Table 1. However, the findings presented require confirmation in prospective studies before broad clinical application.
I recommend the acceptation of the work for publication in Nutrients.